# Modeling the role of the thalamus in resting-state functional connectivity: Nature or structure

**Jesús Cabrera-Álvarez**[1,2]*, **Nina Doorn**[3], **Fernando Maestú**[1,2], **Gianluca Susi**[2,4]

**1** Department of Experimental Psychology, Complutense University of Madrid, Madrid, Spain, **2** Centre for Cognitive and Computational Neuroscience, Madrid, Spain, **3** Department of Clinical Neurophysiology, University of Twente, Enschede, The Netherlands, **4** Department of Structure of Matter, Thermal Physics and Electronics, Complutense University of Madrid, Madrid, Spain

* jescab01@ucm.es

**Data Availability Statement:** The dataset and code used in this study can be found in the GitHub repository: github.com/jescab01/ThalamusInRSFC_2023.

## Abstract

The thalamus is a central brain structure that serves as a relay station for sensory inputs from the periphery to the cortex and regulates cortical arousal. Traditionally, it has been regarded as a passive relay that transmits information between brain regions. However, recent studies have suggested that the thalamus may also play a role in shaping functional connectivity (FC) in a task-based context. Based on this idea, we hypothesized that due to its centrality in the network and its involvement in cortical activation, the thalamus may also contribute to resting-state FC, a key neurological biomarker widely used to characterize brain function in health and disease. To investigate this hypothesis, we constructed ten in-silico brain network models based on neuroimaging data (MEG, MRI, and dwMRI), and simulated them including and excluding the thalamus, and raising the noise into thalamus to represent the afferences related to the reticular activating system (RAS) and the relay of peripheral sensory inputs. We simulated brain activity and compared the resulting FC to their empirical MEG counterparts to evaluate model's performance. Results showed that a parceled version of the thalamus with higher noise, able to drive damped cortical oscillators, enhanced the match to empirical FC. However, with an already active self-oscillatory cortex, no impact on the dynamics was observed when introducing the thalamus. We also demonstrated that the enhanced performance was not related to the structural connectivity of the thalamus, but to its higher noisy inputs. Additionally, we highlighted the relevance of a balanced signal-to-noise ratio in thalamus to allow it to propagate its own dynamics. In conclusion, our study sheds light on the role of the thalamus in shaping brain dynamics and FC in resting-state and allowed us to discuss the general role of criticality in the brain at the meso-scale level.

## Author summary

Synchrony between brain regions is an essential aspect of coordinated brain function and serves as a biomarker of health and disease. The thalamus, due to its centrality and

**Funding:** This research was funded by the Spanish Ministry of Universities through a predoctoral grant to JCA (FPU2019-04251). The funders had no role in study design, data collection and analysis, decision to publish, or preparation of the manuscript.

widespread connectivity with the cortex, is a crucial structure that may contribute to this synchrony by allowing distant brain regions to work together. In this study, we used computational models to investigate the thalamus' role in generating brain synchrony at rest. Our findings suggest that the structural connectivity of the thalamus is not its primary contribution to brain synchrony. Instead, we found that the thalamus plays a critical role in driving cortical activity, and when it is not driving this activity, its impact on brain synchrony is null. Our study provides valuable insights into the thalamocortical network's role in shaping brain dynamics and FC in resting state, laying the groundwork for further research in this area.

## Introduction

In humans, the thalamus is a nut size structure near the center of the brain that relays sensory inputs traveling to the cortex [1], fosters cortico-cortical communication through transthalamic pathways [2, 3], and controls cortical arousal through the reticular activating system (RAS) [4]. To carry out these tasks, it contains three functionally distinct parts [5]: dorsal, ventral, and intralaminar. The dorsal part communicates bidirectionally with the cortex establishing two schemes of information exchange [6, 7] (see Fig 1): first-order relay, in which the thalamus receives subcortical and sensory inputs (i.e., driving inputs), relays them to the cortex through excitatory thalamocortical cells and gets back modulatory feedback from layer 6 pyramidal neurons (i.e., modulatory inputs); and higher order relay, in which the thalamus receives inputs from layer 5 pyramidal cells of a cortical region and relays them to another location in the cortex, creating a transthalamic pathway for cortico-cortical connections [2, 3]. In the ventral part, the reticular nucleus' inhibitory neurons establish connections both with each other and with neurons in the dorsal nuclei, to regulate and foster communication inside the thalamus [8, 9]. The interactions between the dorsal and ventral parts of the thalamus allow for the generation of sustainable oscillations of neural activity, such as delta oscillations, sleep spindles, and slow waves, that may propagate to the cortex and influence its dynamics [10–15].

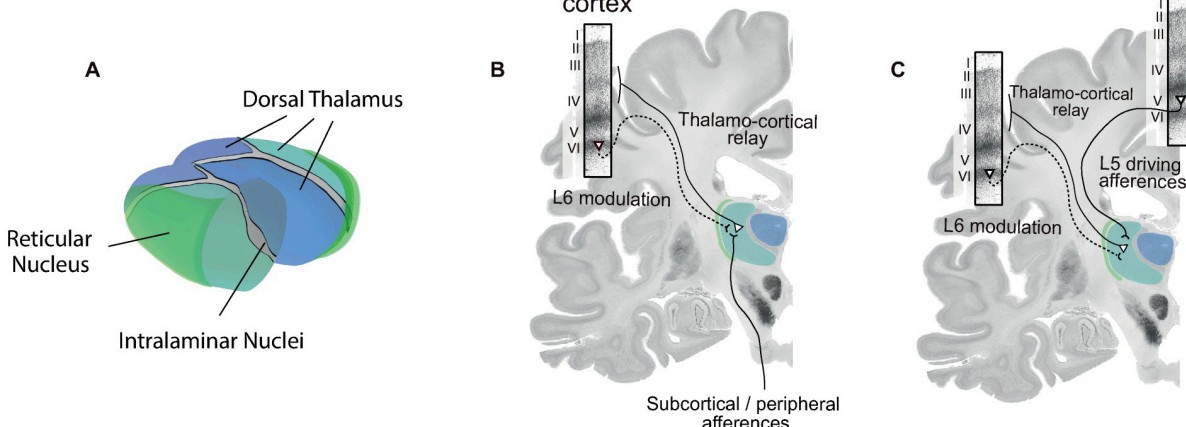

**Fig 1. Main schemes of thalamocortical interaction.** A) Functionally distinct parts of the thalamus including the dorsal thalamus with its anterior, lateral, and medial regions, the reticular nucleus as the main component of the ventral thalamus in green, and the intralaminar nuclei in grey. B) First-order relay scheme in which the subcortical/peripheral afferences are relayed to the cortex generally to layers 3 and 4. C) High-order relay scheme in which afferences from a cortical region are relayed to another. Coronal slices were acquired from the Big Brain project [17] using ebrains platform.

The intralaminar part is involved in the RAS [16] delivering cholinergic and monoaminergic neurotransmission diffusely to the cortex and controlling arousal [4].

Through these pathways, the thalamus connects to a widespread set of cortical regions [18–20], playing a role in different psychological processes such as sleep [10, 21, 22], pain [23, 24], memory and learning [25], attention [26–28], motor and sensory processing [9, 15, 29], and consciousness [30–34]. This role of the thalamus has been classically depicted as a passive relay that transfers information between brain regions [25, 35], however recent findings are challenging this view [20, 25, 36–38]. Specifically, Schmitt et al. [26] showed that the mediodorsal thalamus was amplifying a FC pattern supporting the representation of specific task-related rules. All the thalamocortical mechanisms described above (i.e., the dorsal relay of peripheral sensory inputs, the transthalamic pathways for cortico-cortical communication, and the RAS neurotransmission that activates the cortex) may contribute to define the FC in the brain. Our study is aimed at understanding how.

FC is defined as a correlation between spatially distant neurophysiological signals [39] representing the functional integration of psychological processes in distributed brain networks [40]. Early neuroimaging studies were focused on revealing the activity patterns underlying cognitive processes during task execution, using resting-state as a control condition [41]. However, later findings showed that the brain in resting-state has a rich intrinsic activity [42, 43] related to automatic and unconscious cognitive processing [44, 45]. Since then, resting-state FC (rsFC) has been used to characterize brain function in health and disease [46–48] usually considering it as a static measure. More recently, this approach has been extended to capture the temporal richness of the activity patterns in resting-state through the concept of dynamical FC (dFC; [49, 50] which has been suggested to reflect ongoing cognitive processing and that may be more informative of brain function and dysfunction than the static form [51–54]. Both metrics support the characterization of healthy aging, for which a general decrease in static FC, complemented by a slowing and less complex dFC has been shown and related to changes in cognitive performance [55–58]. Interestingly, some authors have proposed that changes in the thalamocortical network may contribute substantially to the disruptions in FC and cognition during aging [59, 60]. Understanding the mechanisms that underlie and control (d)FC is an important research question that is still undisclosed, especially in aging. Here, we hypothesize that a similar thalamic mechanism that has been shown to be involved in defining FC during task execution [26] might also be active in resting-state.

Computational modeling allows for the generation of in-silico versions of real brains and personalized brain dynamics [61] employing brain network models (BNM). A BNM is based on: a structural connectivity (SC) network derived from diffusion-weighted MRI that captures how brain regions are wired together, and a set of neural mass models (NMM) that reproduce the electrophysiological dynamics of each brain region. A widely studied NMM is the Jansen-Rit (JR; [62]), a biologically-inspired model of a cortical column that implements excitatory and inhibitory subpopulations to produce oscillatory activity. This model shows a bifurcation over a parameter representing the strength of its inputs [63, 64] that will be used in our work to reproduce different modes of the thalamocortical interaction. In a system, a bifurcation occurs when a change in the value of a parameter (i.e., bifurcation parameter) produces a qualitative change in the behavior of the system. For the JR model, the bifurcation separates two different states: a fixed point state, where the model behaves as a damped oscillator (prebifurcation), and a limit cycle state, where the model autonomously oscillates (postbifurcation). These two states turned out to be relevant to understand our findings.

To investigate the potential contribution of the thalamus to rsFC, we built ten in-silico BNMs based on healthy subjects' neuroimaging data (MEG, MRI, dwMRI) using JR NMMs. We simulated them using: 1) three SC versions (i.e., parceled thalamus, pTh; thalamus as a

single node, Th; without thalamus, woTh) to explore the effect of both the parcellation and the mere presence of cortico-cortical transthalamic pathways, and 2) implementing a higher noisy input to the thalamus to reproduce its participation in RAS and the presence of peripheral sensory relays. We compared the simulated FC and dynamical FC (dFC) to their empirical MEG counterparts to evaluate performance. Additionally, we performed further simulations to explore under which conditions the thalamus contributes to the rsFC, including a control experiment using the cortico-cerebellar network instead of the thalamocortical one, and a set of parameter explorations over the intrinsic thalamic oscillatory behavior and the magnitude of the implemented noise. Our results showed that a limited set of driving nodes leading cortical activity was a plausible scenario in rsFC, where the thalamus would play a major role due to its nature: involved in the RAS system, and projecting sensory relays. These results contribute to the understanding of the basic principles of whole-brain function in health and disease, and to enrich the current picture of criticality behavior in the brain.

## Results

### The thalamus impacts rsFC through its afferences

To explore the role of the thalamus in rsFC, we used two features of our in-silico BNMs: its structure, by simulating three different SC versions per subject (pTh, Th, woTh), and the NMMs noisy input by implementing higher than cortex thalamic noise ($\eta_{th}$ = [0.022, 2.2x10$^{-8}$], $\eta_{cx}$ = [2.2x10$^{-8}$]) to represent thalamic RAS system and peripheral first-order relays. We used the coupling parameter ($g$) to scale connectivity weights looking for the best match to empirical rsFC (i.e., the working point), as usual in whole-brain modeling [65–68]. Given that $g$ acts as a bifurcation parameter, the simulated activity could be categorized into two regimes: prebifurcation, in which nodes operate as damped oscillators, and postbifurcation, in which nodes operate as autonomous oscillators. We simulated 60 seconds of brain activity per model and measured FC and dFC in the alpha band to compare to their empirical counterparts using Pearson's correlation ($r_{PLV(\alpha)}$) and Kolmogorov-Smirnov distance (KSD), respectively. We will show statistical comparisons using the best values of those metrics per subject and bifurcation side.

Results on $r_{PLV(\alpha)}$ showed a significant impact for both the structure [F(2, 18) = 191.77, $\eta_g^2 = 0.77$, eps = 0.55, p<0.0001], noise [F(1, 9) = 178.25, $\eta_g^2 = 0.87$, eps = 1, p<0.0001], and their interaction [F(2, 18) = 172.12, $\eta_g^2 = 0.77$, eps = 0.55, p<0.0001] in *prebifurcation*. In contrast, in *postbifurcation*, we did not find significant differences for structure [F(2, 18) = 1.29, $\eta_g^2 = 0.013$, eps = 0.76, p = 0.29] or the interaction [F(2, 18) = 1.74, $\eta_g^2 = 0.004$, eps = 0.65, p = 0.21] while a weak effect was found for noise [F(1, 9) = 5.89, $\eta_g^2 = 0.006$, eps = 1, p = 0.038].

In prebifurcation, implementing high thalamic noise raised significantly $r_{PLV(\alpha)}$ values from close to zero to $r_{PLV(\alpha)} \approx 0.33$ for Th [W = 0, Cohen's d = 5.09, p-corr = 0.005], and $r_{PLV(\alpha)} \approx 0.45$ for pTh [W = 0, Cohen's d = 6.8, p-corr = 0.005] (see Fig 2). The resulting correlation values with high noise differed significantly between the implementations of the thalamic structure [F(2, 18)=186, $\eta_g^2 = 0.87$, eps = 0.53, p-corr<0.0001], and pTh showed a global peak of $r_{PLV(\alpha)}$ that unexpectedly overcame the values observed in postbifurcation in 8 out of 10 subjects (see S1 Fig) although the differences were not statistically significant [W = 18, Cohen's d = 0.142, p = 0.375].

Regarding dFC, the results followed a similar trend in which thalamic structure [F(2, 18) = 119.95, $\eta_g^2 = 0.77$, eps = 0.56, p<0.0001], the thalamic noise [F(1, 9) = 106.3, $\eta_g^2 = 0.73$, eps = 1, p<0.0001] and the interaction [F(2, 18) = 111.66, $\eta_g^2 = 0.77$, eps = 0.55, p<0.0001] were statistically significant factors only in prebifurcation. In that range, high thalamic noise

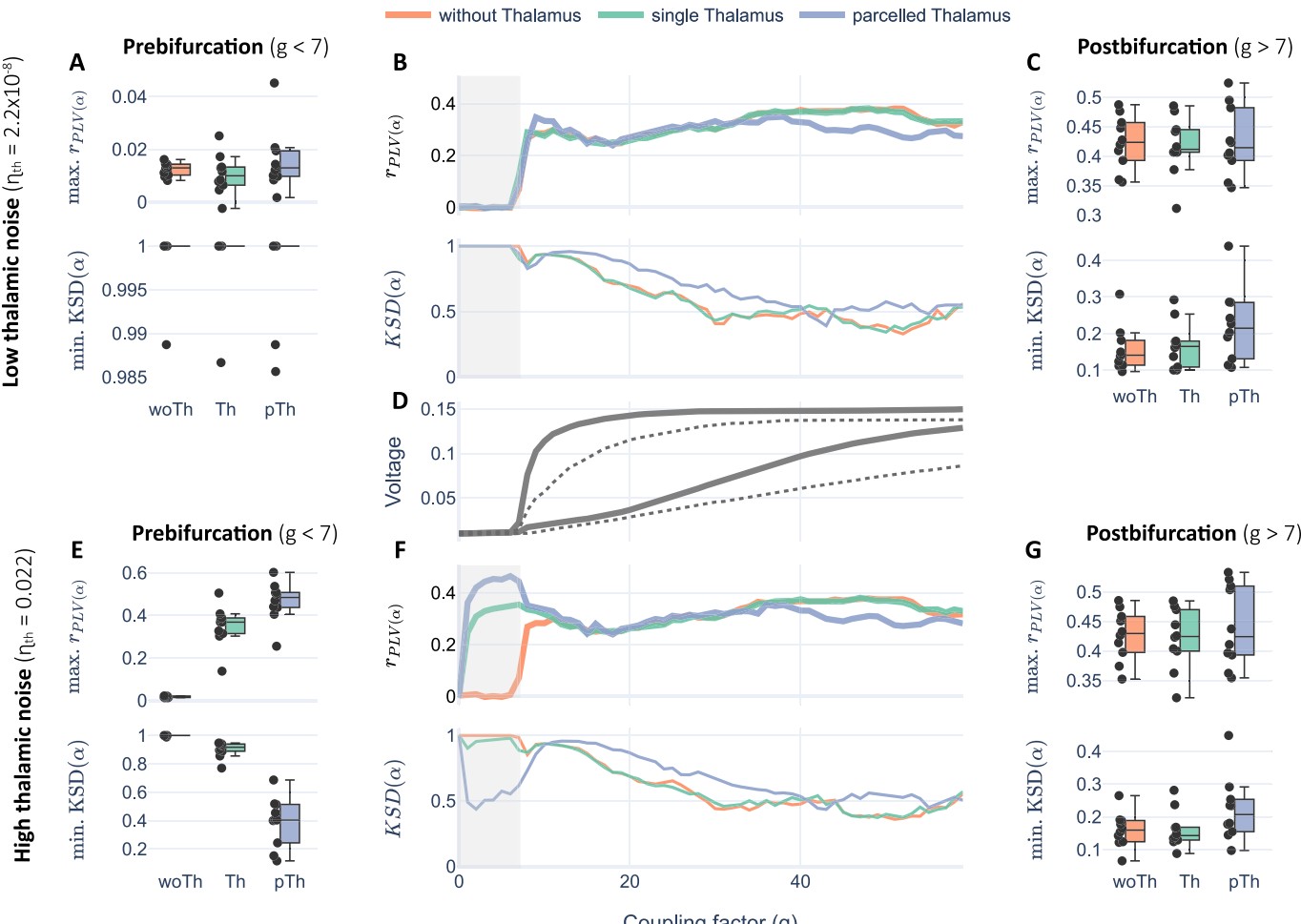

**Fig 2. Thalamocortical experiment.** Simulations for two levels of thalamic noise (low noise in A, B, C; high noise in E, F, G), three implementations of thalamocortical SC (in colours), and exploring the parameter space for coupling factor (g) divided into prebifurcation (g < 7 shadowed regions in B and F) and postbifurcation (g > 7). B, F shows group averaged $r_{PLV(\alpha)}$ and $KSD(\alpha)$ metrics. In the margins, boxplots show maximum values for those metrics per subject in prebifurcation (A, E) and postbifurcation (C, G). D shows the averaged bifurcation diagrams consisting of the maximum and minimum voltages per simulation for the cortex (thick line) and the thalamus (dashed line).

enhanced performance for pTh [W = 0, Cohen's d = 4.89, p-corr = 0.0029] showing a local $KSD(\alpha)$ minimum, and also slightly for Th [W = 0, Cohen's d = 2.5, p-corr = 0.0029] (see Fig 2). Interestingly, we noticed that high values of $KSD(\alpha)$ for woTh and Th were due to opposite underlying dFC distributions. woTh correlations were centered near r = 0 implying that FC matrices in time changed randomly, while Th correlations were centered near r = 1 implying that FC matrices in time were quasistatic (see S2 Fig).

In summary, the inclusion of the thalamus in the model had an impact just in prebifurcation range (in which nodes operate as damped oscillators), and only when implementing a high thalamic noise. For pTh, this condition overcame the performance of any other model. In postbifurcation range (in which nodes autonomously oscillate) the values for $r_{PLV(\alpha)}$ were also high, however, the thalamus did not show a significant impact. The slight differences observed in the boxplots may be related to a higher number of autonomous oscillators trying to impose

their own dynamics that may make it more difficult to establish stable functional interactions between the nodes.

## Structure is not the key: Comparing thalamocortical and cortico-cerebellar networks

We wondered whether the observed improvement of $r_{PLV(\alpha)}$ with high thalamic noise in prebifurcation could be explained by the specific characteristics of the SC pattern of the thalamus. To test this, we performed a control experiment comparing the thalamocortical network to another with similar properties: the cortico-cerebellar network (see Table 1). We built three SC versions per subject: parceled cerebellum (pCer), single node cerebellum (Cer) and without cerebellum (woCer), all of them including the thalamus parceled. We simulated them implementing high noise into the cerebellum to compare the model performance to the thalamocortical network.

We observed that the general trend found with the thalamocortical networks persisted. woCer simulations showed close to zero $r_{PLV(\alpha)}$ values in prebifurcation range with high noise, while Cer and pCer increased significantly their maximum correlations [F(2, 18) = 148.39, $\eta_g^2 = 0.895$, eps = 0.75, p-corr<0.0001] up to similar values obtained with the thalamocortical network (see Fig 3). Moreover, pCer in prebifurcation also resulted in a global maximum in $r_{PLV(\alpha)}$ compared to postbifurcation values. Interestingly, Cer changed the underlying bifurcation diagram of the model, moving it toward higher values of coupling. This was reflected by the peak of $r_{PLV(\alpha)}$ in higher g values.

These results suggest that the specific thalamocortical SC pattern is not a major determinant of the thalamic contribution to rsFC.

## Brain dynamics underlying each scenario

To understand the dynamics that underlie the observed values of $r_{PLV(\alpha)}$ and KSD, we extracted a simulation sample per model condition with pTh (i.e., high/low thalamic noise, and pre- / post-bifurcation; see Fig 4).

The dynamics in the prebifurcation range with high and low noise showed 1/f pink noise pattern. This is the result of a damped JR node processing a Gaussian noise as shown in previous literature [69, 70]. High and low noise conditions in prebifurcation could be differentiated by their spectral powers and by the differences in FC matrices: with low noise, nodes are not powerful enough to interact, producing a functional disconnection that was captured by the FC and dFC matrices. In sharp contrast, in postbifurcation, nodes were self-oscillating in alpha frequency around 10Hz. Note the similarity between high and low noise to the thalamus in postbifurcation range, supporting the results reported in previous sections.

**Table 1. Network features for the thalamus and the cerebellum.**

|  | Degree | Node strength | Betweenness | Path Length |
|---|---|---|---|---|
| Global average | 0.827 | 0.231 | 0.00119 | 1.165 |
| Thalamus average | 0.851 | 0.111 | 0.00125 | 1.141 |
| Cerebellum average | 0.883 | 0.224 | 0.00132 | 1.110 |

Averaged network metrics for the thalamus, the cerebellum, and the global average of all regions. Similar relation to average was observed between regions in all metrics (degree, betweenness, and path length) except for node strength. For more details on the network analysis, see S1 Table.

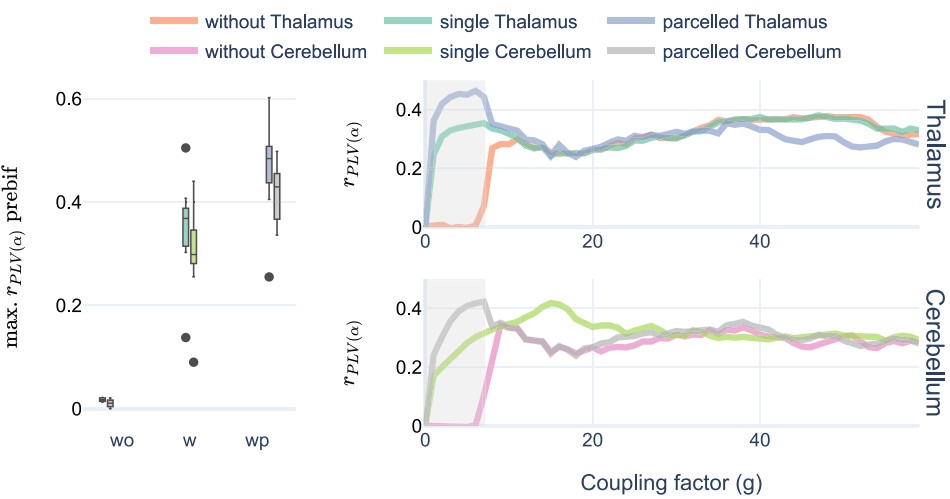

**Fig 3. Cortico-cerebellar control experiment.** Comparing the simulations implementing high noise in thalamus and in cerebellum with three versions of their structure. Left, boxplot showing maximum $r_{PLV(\alpha)}$ values per subject in the prebifurcation space for both experiments. Right, lineplots conveying averaged values of $r_{PLV(\alpha)}$ per value of coupling factor, and SC version. Shadowed areas covering the prebifurcation range.

## Thalamic alpha propagates to the cortex within a balanced SNR

The best-performing thalamocortical model was the one in prebifurcation that integrated high thalamic noise and pTh. Looking at its underlying dynamics, we observed that the spectrum was showing a 1/f shape. As we are trying to reproduce MEG FC, in which a predominant alpha frequency is usually observed, we wondered whether a spectral change towards alpha would impact the model performance. In this section, we manipulated the oscillatory frequency of the thalamus making it surpass bifurcation and self-oscillate in alpha by varying its average input, $p_{th}$. These simulations were performed with pTh structure and $\eta_{th}$ = 0.022.

Simulations rising $p_{th}$ in prebifurcation (g<7), resulted in a transition of thalamic nodes from the noisy 1/f pink noise spectra to an alpha oscillation (see Fig 5, FFT peak), passing first through the slow and high amplitude limit cycle of the JR model [63, 64] at $p_{th}$ = [0.11— 0.13] (see the dark orange spot in Fig 5 SNR). In this transition along the bifurcation, $r_{PLV(\alpha)}$ lowered down right after the high power and slow limit cycle ($p_{th}$ > 0.13). This could be due to a high SNR $\approx$ 6 producing hypersynchronization (mean PLV$\approx$0.7; see Fig 5 mean PLV). However, we did not find this phenomenon with the slow limit cycle (PLV($\alpha$)mean $\approx$0.5) even though it showed a higher SNR. This might be explained by the alpha band filtering that leaves out of the analysis the potentially hypersynchronizing oscillations of the slow limit cycle.

Interestingly, the changes in thalamic activity indirectly increased the inter-regional inputs to cortical nodes, making some of them pass bifurcation at g$\approx$3 and g$\approx$6 (see the horizontal blue lines in Fig 5, FFT peak). These nodes produced a further rise in $r_{PLV(\alpha)}$ (see the horizontal red line in Fig 5, $r_{PLV(\alpha)}$) suggesting two important things for prebifurcation simulations: 1) that every node is a potential contributor of a general driving mechanism that we have located in the thalamus (through a high noise), and 2) that the number of drivers participating in that mechanism matters.

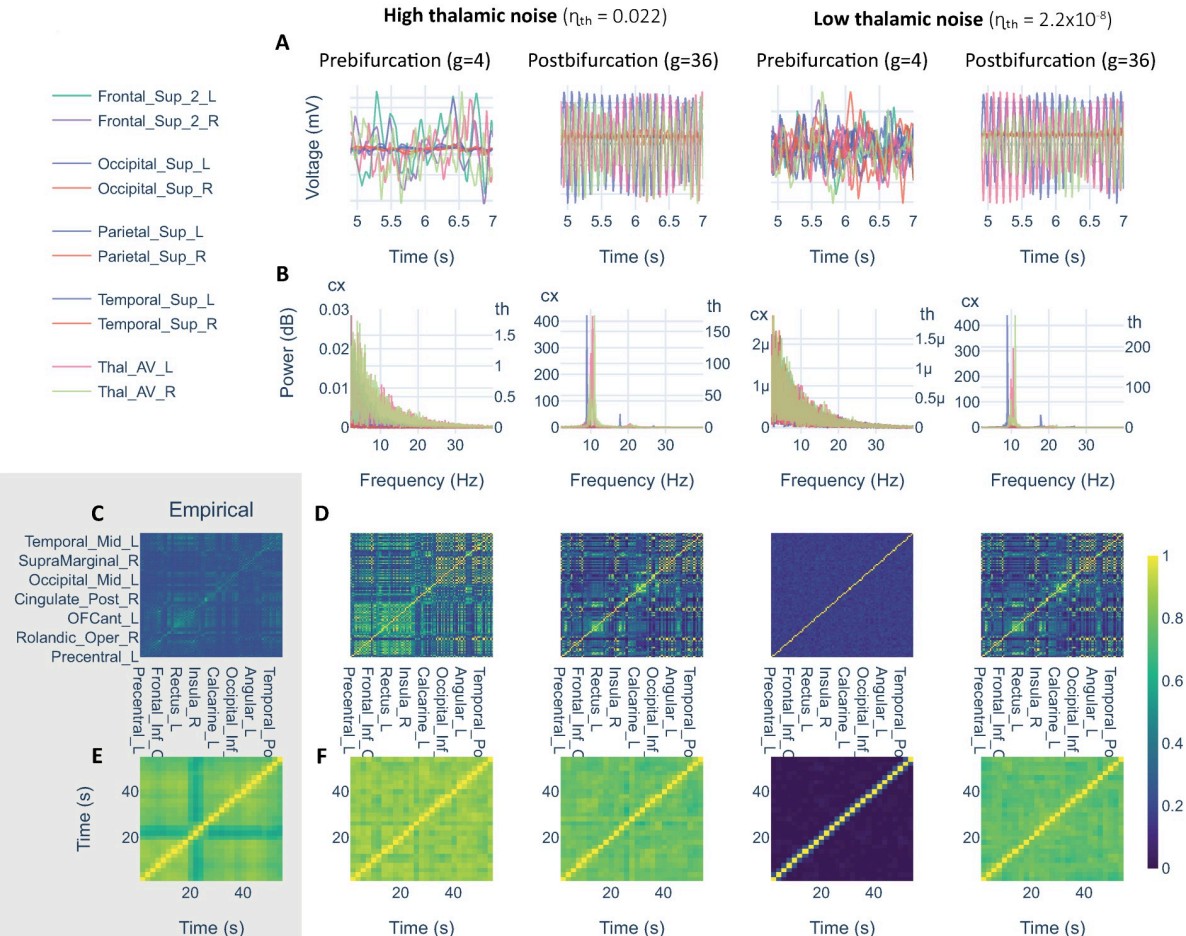

**Fig 4. Simulation samples for one subject with parceled thalamus.** In columns, simulation samples combining low/high thalamic noise with coupling values from pre/post- bifurcation. We selected g = 4 to simulate prebifurcation and g = 36 to simulate postbifurcation. First two rows show samples from the simulated signals (A), and their corresponding spectra (B). Last two rows showing PLV($\alpha$) (C, D) and dFC ($\alpha$) (E, F). C and E showing empirical references for PLV($\alpha$) and dFC($\alpha$), respectively.

In the range of $p_{th}$ = [0.13–0.3], where we observed alpha oscillations, $r_{PLV(\alpha)}$ decreased. We wondered whether this effect could be related to the rise in SNR after setting the thalamus to oscillate in the alpha band. Therefore, we fixed $p_{th}$ = 0.15, and we varied the noisy input to the thalamus ($\eta_{th}$) to explore its impact on model performance. We found an optimal balance for SNR at $\eta_{th}$ = [0.05–0.15] in which the noisy inputs to the thalamus were enough to avoid hypersynchronization (see Fig 6 mean PLV) and low enough to maintain the intrinsic dynamics produced by the thalamus (i.e., alpha oscillations, see Fig 6 FFT peak). Further increases in noise would replace progressively the alpha oscillatory behavior by a 1/f spectra without reducing $r_{PLV(\alpha)}$.

From these observations, it could be thought that a general rise of noise in the model (i.e., to all nodes) would lead to better performance, however, in our modeling framework this is only true when the noise is implemented into a limited number of nodes. Independent noise into all cortical nodes is not linked to an enhancement of $r_{PLV(\alpha)}$ (see S3 Fig).

From these parameter explorations, we extracted four additional models of interest to explore their underlying dynamics (see Fig 7). Two of them related to the exploration of $p_{th}$:

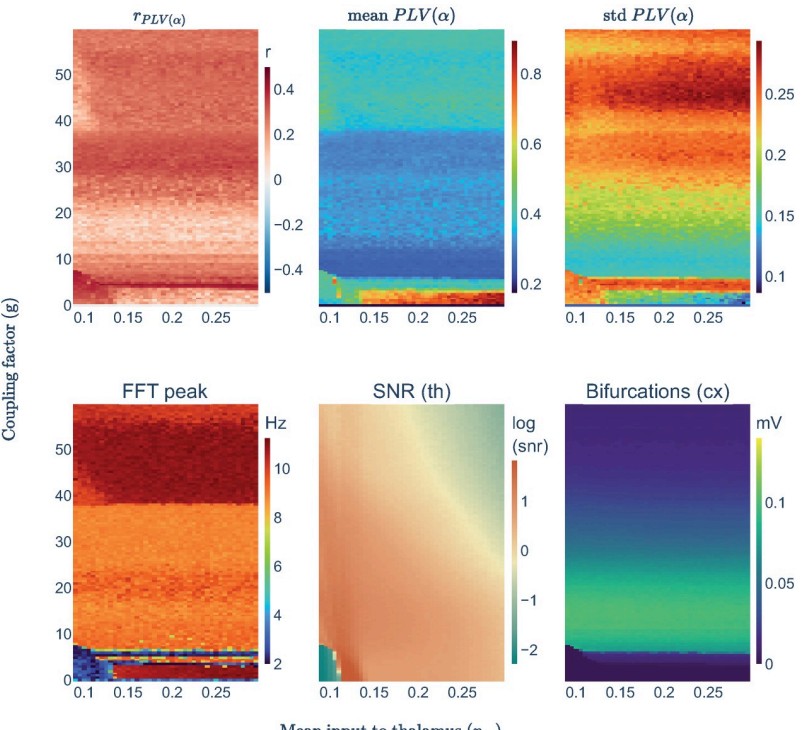

**Fig 5. Parameter space explorations for p_th with high thalamic noise.** Sets of three simulations averaged for subject one and parceled thalamus. Heatmaps showing different metrics from the same each simulation: the empirical-simulated correlation of PLV in alpha band ($r_{PLV(\alpha)}$), the simulated mean and std of the $PLV(\alpha)$ values, the frequency peak of the nodes' averaged spectra (FFT peak), the signal to noise ratio in thalamic nodes computed as the amplitude of simulated signals divided by the standard deviation of the Gaussian noise used for the thalamus (SNR(th)), and the bifurcation of cortical signals using the averaged maximum-minimum signals' voltage.

one for the hypersynchrony situation ($p_{th} = 0.15$, $\eta_{th} = 0.022$), another for the slow JR limit cycle ($p_{th} = 0.12$, $\eta_{th} = 0.022$); and another two regarding the exploration of SNR: one for the optimal SNR ($p_{th} = 0.15$, $\eta_{th} = 0.09$), and another for a higher noise than the optimal range ($p_{th} = 0.15$, $\eta_{th} = 0.5$). Fig 7 shows the underlying dynamics for each of the additional model scenarios simulated with g = 2.

## Discussion

Understanding thalamocortical networks is crucial for unraveling the complex dynamics of the human brain. These networks are essential for transmitting sensory information from the periphery to the cortex and regulating cortical arousal, which are fundamental processes for perception, attention, and cognition. In this study, we aimed to gain a deeper understanding of the role of the thalamus in rsFC by utilizing computational brain models. Our experiments focused on testing two key features of thalamocortical networks: the presence of thalamic afferences related to the RAS and first-order sensory relays, and the structure of the thalamus. Interestingly, we found that only when we raised thalamic noisy inputs to represent the RAS and peripheral afferences activating a damped cortex, its presence affected the simulated dynamics. To validate our findings, we performed a control experiment using the cerebro-cerebellar network and showed that implementing high noise into the cerebellum could replicate the results observed in the thalamocortical network. Finally, we explored how the oscillatory

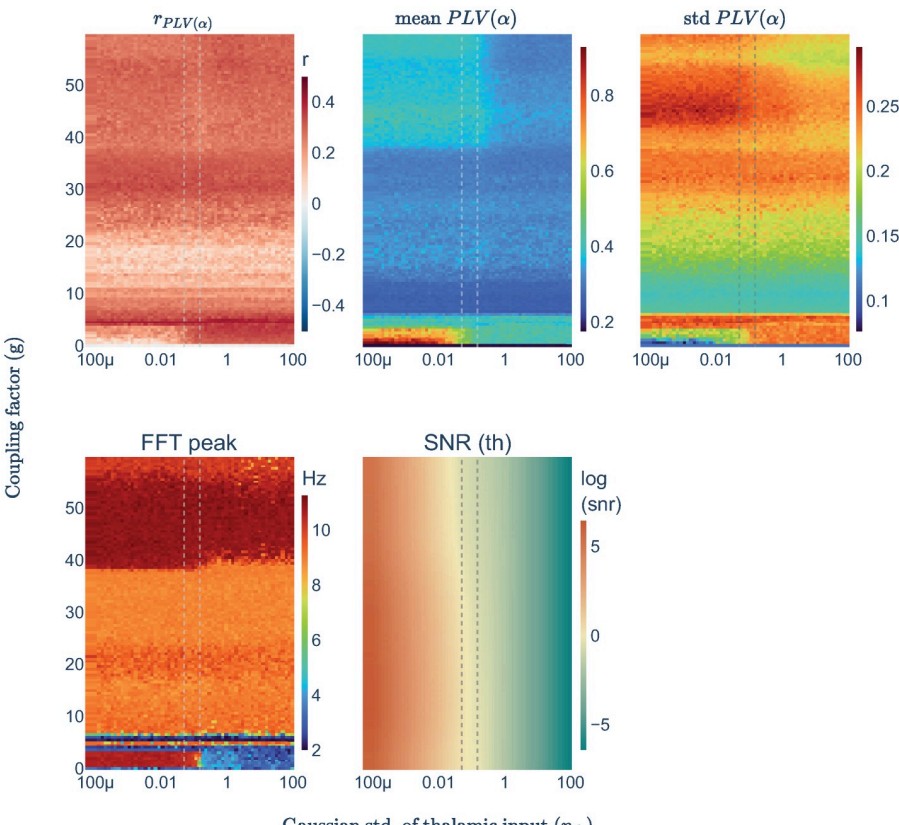

**Fig 6. Parameter space explorations over $\eta_{th}$ to balance SNR in the thalamus.** Sets of three simulations averaged for subject one and parceled thalamus. Heatmaps showing different metrics from the same each simulation: the empirical-simulated correlation of PLV in alpha band ($r_{PLV(\alpha)}$), the simulated mean and std of the $PLV(\alpha)$ values, the frequency peak of the nodes' averaged spectra (FFT peak), the signal to noise ratio in thalamic nodes computed as the amplitude of simulated signals divided by the standard deviation of the Gaussian noise used for the thalamus (SNR(th)), and the bifurcation of cortical signals using the averaged maximum-minimum signals' voltage. Vertical dashed lines define the optimal SNR range.

behavior of thalamic nodes, specifically their frequency and SNR, could shape the emergent rsFC. Our study provides novel insights into the role of thalamocortical networks in shaping brain dynamics and highlights the relevance of balanced SNR activity for the propagation of alpha rhythms from the thalamus.

We expected that introducing the thalamus in our simulations would have generated a difference in model performance in postbifurcation, where the best correlations are usually found. However, this was not the case, as implementing the thalamus in our model only affected results in prebifurcation and when introducing a high thalamic noise to represent afferences from RAS system and peripheral sensory relays. In prebifurcation, nodes are operating as damped oscillators tending to relaxation at a fixed point. When we introduced high noise in the thalamus, its afferences to cortex rose cortical activation levels allowing for functional interactions. In this situation, the thalamus is driving cortical activation. Additionally, the best model performance was observed within this parametrization and including the thalamus with its dorsal nuclei divisions (pTh). More importantly, the postbifurcation range has been often associated with a state of generalized epileptic seizure [71, 72] due to its highly synchronized intra-node oscillatory activity (see the spectra at postbifurcation in Fig 4B) that is

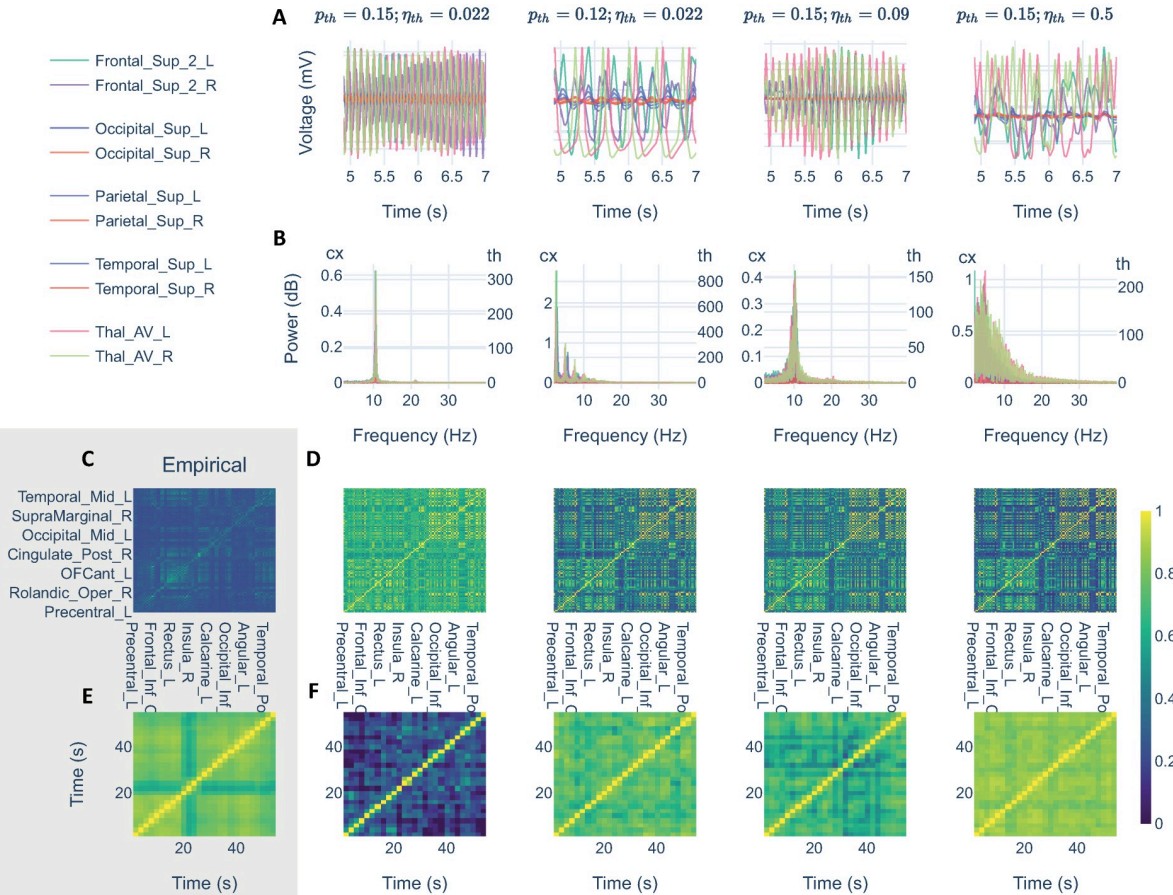

**Fig 7. Complementary simulation samples.** Simulation samples derived from the parameter explorations with parceled thalamus and cortical nodes in prebifurcation (g = 2). In columns, simulation samples in which the thalamus drives cortical activity with different levels of noise ($\eta_{th}$ = [0.022, 0.09, 0.5]) and at two different points of the thalamic bifurcation determined by $p_{th}$ = [0.12, 0.15], the former corresponds to the slow limit cycle of JR. First two rows show samples from the simulated signals (A), and their corresponding spectra (B). Last two rows showing PLV($\alpha$) (C, D) and dFC($\alpha$) (E, F). C and E showing empirical references for PLV($\alpha$) and dFC($\alpha$), respectively.

not the empirical target of this study (i.e., MEG resting state). Taken together, these ideas led us to focus on the prebifurcation range.

A further control experiment demonstrated that this mechanism was not directly related to thalamic SC. This was ascertained by embedding the driver mechanism into another hub-like region (i.e., cerebellum) and obtaining equivalent results for the match to empirical rsFC. This would mean that in our model, where all neural masses are functionally equal, any node could theoretically play the role of the thalamus. In this line, during parameter explorations, some isolated cortical nodes passing bifurcation would contribute to enhance performance by being part of that driving mechanism. This would add up to the observation that parceled structures (i.e., thalamus and cerebellum) performed better than their single node versions, indicating that the number of driver nodes matters. In further research, it should be explored whether there is a computational optimum number of drivers to reproduce rsFC that could be subject or session-specific, and whether these differences may be linked to differences in brain and cognitive functioning.

Taken together, our main conclusion is that *a limited set of driving nodes is likely to underlie the dynamics of rsFC*. We believe that those drivers might be linked both to the regions participating in the RAS system (Intralaminar Thalamus, Raphe Nuclei, and Locus Coeruleus) [4, 16] and to the dorsal nuclei of the thalamus that are implicated in the relay of sensory information and have also been tightly linked to the generation of oscillatory behavior in the cortex in slow waves [73–76]. This would support the view of the thalamus as a driver and controller of cortical dynamics [6, 26, 77–79].

Further explorations on the spectral characteristics of the drivers showed that the thalamus could propagate its own intrinsic alpha dynamics when a balanced SNR was achieved. This feature represents the interaction between thalamocortical pacemaker neurons [14] and peripheral sensory inputs reaching the system and provoking event-related desynchronization [80, 81]. The model showed a good performance for reproducing empirical rsFC in that optimal range and with additional noise, generating a 1/f spectra. However, lower levels of noise with an alpha-oscillating thalamus reduced performance and led to a hypersynchronization situation transmitted from the thalamus to the cortex, generating an epileptic-like dynamic [82, 83]. In line with these results, the thalamus has been proposed to be involved in the onset of temporal lobe epileptic seizures, transmitting more regular patterns of activity to the hippocampus [84, 85].

From the computational perspective, previous work has shown that BNMs may reproduce better empirical rsFC when the models operate at the edge of bifurcation [67], the critical point. At this point, noisy excursions or the effect of nodes' interaction can lead the masses to behave in any of the two regimes separated by the critical point. This phenomenon, referred to as criticality [66], has been proposed to enhance the capacity of brain systems to convey information [86]. However, in our study, we showed an equivalent performance both at the edge of bifurcation and over the whole prebifurcation range. This contrast might be explained by the different cortical and thalamic parametrization implemented in our nodes. Our thalamic driving nodes feed the cortex, leading the dynamics. In the cited study [67], the nodes that randomly switch between states would represent the same driving mechanism as in our model. Interestingly though, our approach would suggest that criticality is not a necessary feature in the dynamics of a resting brain at the mesoscale level.

Many studies in the field have explicitly [87–89] or implicitly [66, 90–92] excluded subcortical regions from their BNMs. This could be due to the technical limitations of recording deep brain signals and/or to the complexity of reconstructing SC schemes for small, deep crossing-fibers regions. But, more importantly, it could be due to the unknown role that these regions may play in shaping simulated whole brain dynamics. Some studies have attempted to unravel these mechanisms, showing the importance of the cerebellum for brain dynamics [68], the relevance of cortico-subcortical interaction for shaping dynamical functional connectivity [93] and the relevance of neurotransmission [65, 94]. Additionally, other studies are paving the way towards multiscale computational models in which subcortical areas are implemented with a further spatiotemporal level of detail [95–98]. These approaches could be an interesting path to extend our knowledge regarding the potential role of the thalamus (and other activating brain regions) in whole brain simulations.

In conclusion, our study provides novel insights into the role of thalamocortical networks in shaping brain dynamics. We demonstrate that a limited set of driving nodes leading cortical activation may better describe resting-state activity. The thalamus would be a relevant part in this mechanism due to its participation in the RAS system and through its peripheral sensory relays being delivered from its multiple dorsal nuclei. In this type of architecture, driving

nodes might show a balanced SNR to avoid hypersynchronization in the network. Although it is still debated whether the thalamus has an active role in cognition [6, 25, 99], our study strongly suggests its active participation in driving cortical dynamics and shaping FC in resting-state. These findings may contribute to a better understanding of brain function and dysfunction, fostering the development of new therapeutic approaches targeting thalamocortical circuits.

## Materials and methods

### Empirical dataset

MRI (T1 and DWI) scans and MEG recordings were acquired from 10 healthy participants in resting-state, with ages between 62 and 77 years old (mean 69, sd 4.17, 3 males, 7 females) from a dataset owned by the Centre of Cognitive and Computational Neuroscience, UCM, Madrid.

MRI-T1 scans were recorded in a General Electric 1.5 Tesla magnetic resonance scanner, using a high-resolution antenna and a homogenization PURE filter (fast spoiled gradient echo sequence, with parameters: repetition time/echo time/inversion time = 11.2/4.2/450 ms; flip angle = 12˚; slice thickness = 1 mm, 256×256 matrix, and field of view = 256 mm).

Diffusion-weighted images (dw-MRI) were acquired with a single-shot echo-planar imaging sequence with the parameters: echo time/repetition time = 96.1/12,000 ms; NEX 3 for increasing the SNR; slice thickness = 2.4 mm, 128×128 matrix, and field of view = 30.7 cm yielding an isotropic voxel of 2.4 mm; 1 image with no diffusion sensitization (i.e., T2-weighted b0 images) and 25 dw-MRI (b = 900 s/mm2).

MEG recordings were acquired with an Elekta-Neuromag MEG system with 306 channels at 1000Hz sampling frequency and an online band-pass filtered between 0.1 and 330Hz. MEG protocol consisted of 5 min resting-state eyes closed.

All participants provided informed consent.

### Functional connectivity

MEG recordings were preprocessed offline using the spatiotemporal signal space separation (tSSS) filtering algorithm [100], embedded in the Maxfilter Software v2.2 (correlation limit of 0.9 and correlation window of 10 seconds), to eliminate magnetic noise and compensate for head movements during the recording. Continuous MEG data were preprocessed using the Fieldtrip Toolbox [101], where an independent component-based algorithm was applied to remove the effects of ocular and cardiac signals from the data, together with external noise.

Source reconstruction was performed using the software Brainstorm [102], anatomically informed by the MRI scans of each subject. We employed the minimum norm estimates method [103], with the *constrained dipoles* variant, by which the current dipoles are oriented normally to the cortical surface, to model the orientation of the macrocolumns of pyramidal neurons, perpendicular to the cortex [104]

Source-space signals were then filtered in the alpha band (8–12 Hz) to calculate FC between the time series using the Phase Locking Value (PLV($\alpha$), [105]), and the resulting matrices were averaged into the AAL2 parcellation scheme [106]. We restricted the analysis to 1) cortical regions, to avoid the limitations of MEG recordings regarding deep brain signals [107]; and 2) the alpha band, for computational simplicity and being aware that it dominates MEG resting-state recordings. In addition, we computed dynamical functional connectivity matrices by extracting PLV($\alpha$) on consecutive intervals of 4 seconds of length with the sliding window approach and 50% of overlapping [108], and evaluating the correlation between these PLV($\alpha$) matrices.

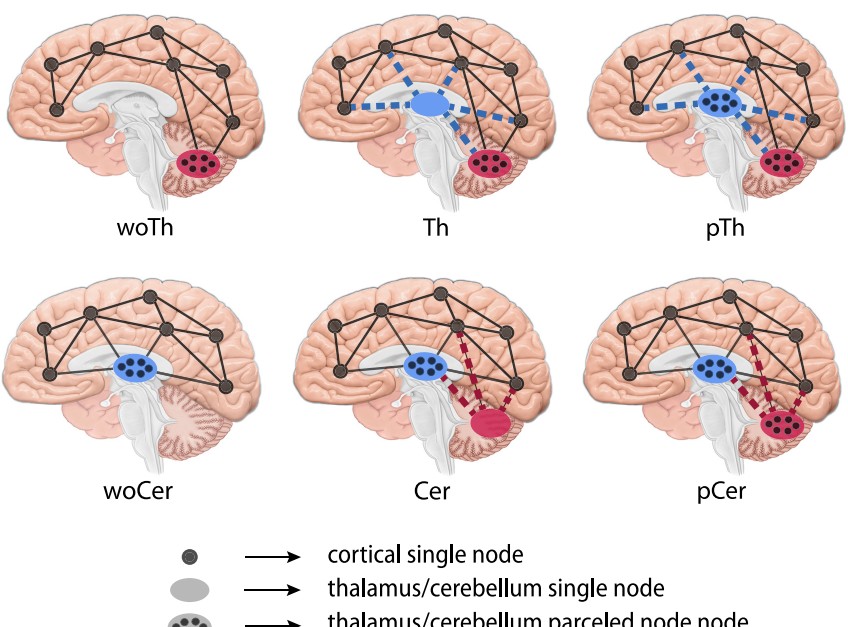

**Fig 8. SC versions.** First line shows the SC versions for the thalamocortical experiment: woTh, Th, pTh. The second line shows the SC versions for the cortico-cerebellar control experiment. Dashed lines representing driver connections.

## Structural connectivity

Diffusion-weighted images were processed using DSI Studio (http://dsi-studio.labsolver.org). The quality of the images was checked before fiber tracking and corrected for motion artifacts, eddy currents, and phase distortions. Then, tensor metrics were calculated. To improve reproducibility, we used a deterministic fiber tracking algorithm with augmented tracking strategies [109–111]. The whole brain volume was used as seeding region. Both the anisotropy and angular thresholds were randomly selected (the latter, from 15 degrees to 90 degrees). The step size was randomly selected from 0.5 voxels to 1.5 voxels. A total of 5 million seeds were placed and tracks with lengths shorter than 15 or longer than 180 mm were discarded.

To explore the impact of including/excluding the thalamus in simulations, we performed a first experiment comparing three different structural connectivity (SC) versions of each subject brains': woTh, Th, and pTh. pTh consists of a brain network with 148 regions extracted from AAL3 atlas from which we kept the thalamic parcellation and, removed and merged the other areas to make it comparable to AAL2 scheme; Th consists of the 120 regions from AAL2; same for woTh in which we removed thalamic nuclei (118 regions). These three versions of the structural connectomes are represented in Fig 8 and included the cerebellum parceled into its nuclei. A list with all ROIs included can be found in S1 Table. Two connectivity matrices were calculated per SC version: counting the number of tracts connecting (i.e., passing through) each pair of brain regions, and the average length of those tracts.

As a control experiment, we applied the same process to the cerebellum as it is a brain region with similar network characteristics (see Table 1 and S1 Table), and it can also be modeled as a parceled structure and a single node. We extracted three SC versions per brain using AAL3 atlas: parceled cerebellum (pCer), cerebellum as a single node (Cer), and without cerebellum (woCer). The thalamus was modeled as parceled through all these versions, and therefore the resulting SCs were composed of 148, 122 and 120 regions, respectively.

## Brain network model

SC matrices served as the skeleton for the BNMs implemented in TVB [112] where regional signals were simulated using JR NMMs [62]. This is a biologically inspired model of a cortical column capable of reproducing alpha oscillations through a system of second-order coupled differential equations:

$$\dot{y}_{0_i}(t) = y_{3_i}(t) \tag{1}$$

$$\dot{y}_{1_i}(t) = y_{4_i}(t) \tag{2}$$

$$\dot{y}_{2_i}(t) = y_{5_i}(t) \tag{3}$$

$$\dot{y}_{3_i}(t) = AaS[y_{1_i}(t) - y_{2_i}(t)] - 2ay_{3_i}(t) - a^2 y_{0_i}(t) \tag{4}$$

$$\dot{y}_{4_i}(t) = Aa(input(t) + C_2 S[C_1 y_{0_i}(t)]) - 2ay_{4_i}(t) - a^2 y_{1_i}(t) \tag{5}$$

$$\dot{y}_{5_i}(t) = Bb(C_4 S[C_3 y_{0_i}(t)]) - 2by_{5_i}(t) - b^2 y_{2_i}(t) \tag{6}$$

Where:

$$S[v] = \frac{(2 \cdot v_{\max})}{1 + \exp^{r(v_0 - v)}} \tag{7}$$

The inter-regional communication introduces heterogeneity in terms of connection strength $w_{ji}$, and conduction delays $d_{ji}$ (i.e., tract length / conduction speed) between nodes i and j, where:

$$input(t) = p_i + \eta_i(t) + g\sum_{j=1}^{n} w_{ji} \cdot S[y_{1_j}(t - d_{ji}) - y_{2_j}(t - d_{ji})] \tag{8}$$

It represents the electrophysiological activity (in voltage) from three subpopulations of neurons: pyramidal neurons ($y_0$), excitatory interneurons ($y_1$), and inhibitory interneurons ($y_2$). These subpopulations are interconnected (Fig 9) and integrate external inputs from other

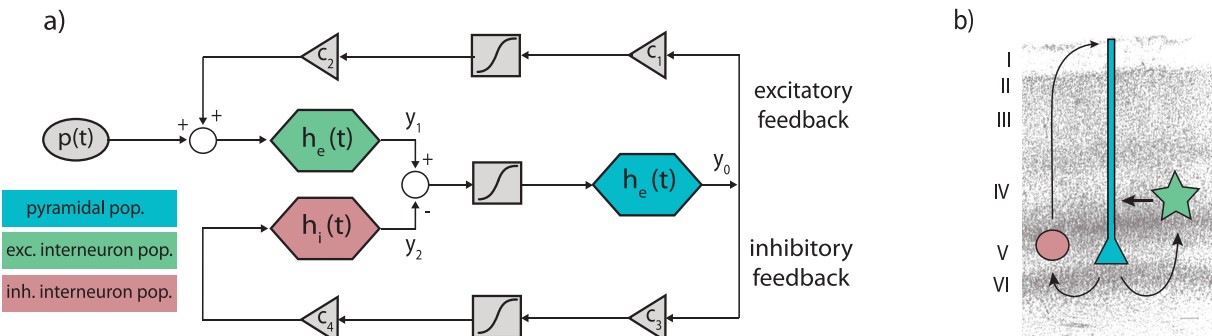

**Fig 9. JR model of a cortical column.** a) Block diagram depicting JR operators and modules where each color is associated with a different neural population: pyramidal (cyan), excitatory interneurons (green) and inhibitory interneuron (red). b) Histological contextualization of the cortical layers. Modified from [62, 90].

**Table 2. JR parameters used in simulations.**

| Parameter | Value | Unit | Description |
|---|---|---|---|
| A | 3.25 | mV | Average excitatory synaptic gain |
| B | 22 | mV | Average inhibitory synaptic gain |
| a | 0.1 | ms$^{-1}$ | Time Constant of excitatory PSP |
| b | 0.05 | ms$^{-1}$ | Time Constant of inhibitory PSP |
| C1 | 135 | | Average synaptic contacts: pyramidals to excitatory interneurons |
| C2 | 108 | | Average synaptic contacts: excitatory interneurons to pyramidals |
| C3 | 33.75 | | Average synaptic contacts: pyramidals to inhibitory interneurons |
| C4 | 33.75 | | Average synaptic contacts: inhibitory interneurons to pyramidals |
| $v_{max}$ | 0.0025 | ms$^{-1}$ | Half the maximum firing rate |
| r | 0.56 | mV$^{-1}$ | Slope of the presynaptic function at $v_0$ |
| $v_0$ | 6 | mV | Potential when half the maximum firing rate is achieved |
| p | variable | ms$^{-1}$ | Mean of random Gaussian intrinsic input |
| $\eta$ | variable | ms$^{-1}$ | Standard deviation of random Gaussian intrinsic input (noise) |
| g | variable | | Coupling factor for inter-regional communication - multiplier of weights - |
| s | 15 | mm/ms | Conduction speed for inter-regional communication |

Unless otherwise stated, we used default values for parameters $p = 0.09$ and $\eta = 2.2 \times 10^{-8}$. Note that along the study, we introduce bimodalities in those parameters $p_{th}$ and $\eta_{th}$)

cortical columns. The communication is implemented in terms of firing rate (Eqs 1 to 6) and a sigmoidal function (Eq 7) stands for the conversion from voltage to firing rate.

The input represents two main drivers of activity in the NMMs: inter-regional communication and intrinsic input. The former consists of the signal transmission between nodes through the SC of the brain in which weights are linearly scaled by a global coupling factor $g$, and tract lengths are divided by conduction speed to define $d_{ji}$. Conduction speed was set to 15 m/s given the low impact shown in previous parameter space explorations done in this project (see S5 Fig). The latter is defined by a Gaussian noise with $p$ mean and $\eta$ std. Parameter values are described in Table 2.

The JR model shows two supercritical hopf bifurcations for the parameter $p$ [64]. When JR NMMs are implemented in a connected network, the parameter $g$ scales the inter-regional input to nodes, becoming a bifurcation parameter. We used the first bifurcation to separate two NMM's behaviors (Fig 10): damped oscillator in the *prebifurcation* range where nodes

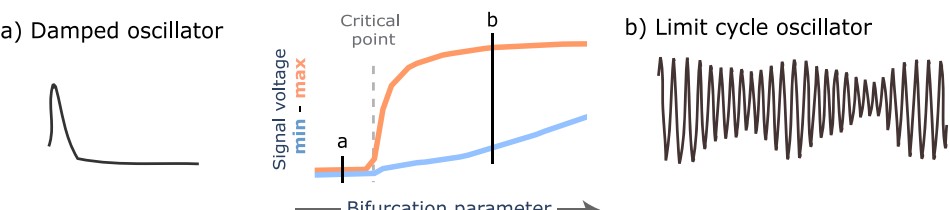

**Fig 10. The bifurcation separates two states.** In the center, a bifurcation diagram shows the minimum and maximum voltage for each value of the bifurcation parameter. At the critical point (dashed line), the bifurcation occurs and separates two states of the system: a) damped oscillator, whose activity tends to decay to a fixed point; and b) limit cycle oscillator, whose activity is a self-sustained oscillation.

## Simulations

In the first experiment, simulations were performed varying two parameters: the standard deviation of the input to the thalamus (i.e., noise, $\eta_{th}$ = [0.022, 2.2x10$^{-8}$]) representing the presence/absence of subcortical and peripheral inputs; and the thalamic structure (woTh, Th, pTh). The higher noise level was determined following previous research [12], while the lower was chosen to avoid flat signals in the fixed state of the JR model. Additionally, we explored the parameter space for coupling factor (g=[0–60]). These models were simulated with the parameter $p$ set to 0.09. We performed three simulations of 60 seconds (removing the initial 4 seconds to avoid transients) per model and computed two metrics: the Pearson's correlation coefficient between the vectorized upper triangular matrices of the simulated and empirical FC (i.e., $r_{PLV(\alpha)}$); and the KSD($\alpha$) between the distributions of correlations in the dFC matrices (empirical and simulated). The same configuration was used in the control experiment with the cerebellum.

For further explorations, we simulated for 10 seconds (omitting the initial 2 seconds to avoid transients) different ranges of the mean intrinsic input to the thalamus $p_{th}$) and its standard deviation ($\eta_{th}$). Besides the $r_{PLV(\alpha)}$ and KSD($\alpha$) metrics, we show 1) *bifurcation diagrams* capturing the averaged maximum and minimum signal's voltage per simulated ROI at each point in the parameter space. In the case of exploring 2 parameters at the same time (e.g., $g$ and $p_{th}$), bifurcations are presented in heatmaps conveying information about the difference between the maximum and minimum signal voltage; 2) *signal-to-noise ratio* (SNR) in the thalamus that is computed by dividing the amplitude of simulated signals by the standard deviation of the Gaussian noise used for the thalamus; 3) *relative power* between cortex and thalamus calculated by dividing the averaged area of cortical spectra by the averaged area of thalamic spectra.

## Statistics

We averaged the results of the 3 sets of simulations (i.e., repetitions) and performed statistical analysis for the group of 10 subjects. For the first experiment, we considered the maximum $r_{PLV(\alpha)}$ and minimum KSD($\alpha$) per subject, thalamic SC version and scenario. The effects of thalamic SC version and noise levels were evaluated using four two-way repeated measures ANOVA: two comparing maximum $r_{PLV(\alpha)}$ in prebifurcation and postbifurcation; and another two comparing minimum KSD; after checking for the statistical assumptions of normality (Shapiro's test) and sphericity (Mauchly's test). Pairwise comparisons for thalamic structure and thalamic noise were evaluated using Wilcoxon test, correcting for multiple comparisons using FDR Benjamini-Hochberg method. The same procedure was applied in the control experiment with the cerebro-cerebellar network to maximum $r_{PLV(\alpha)}$ comparisons.

## Supporting information

**S1 Fig. Lineplots showing $r_{PLV(\alpha)}$, KSD($\alpha$) and bifurcations per subject and SC version.** The global behavior in $r_{PLV}$(first column) was similar for every subject. Note slight differences for subject 2 and subject 8 in which the bifurcation does not match the highest $r_{PLV}$ value. (TIF)

**S2 Fig. Empirical and simulated distributions of Pearson's correlation values in dFC matrices.** Simulations were performed for subject 1 with high thalamic noise ($\eta_{th}$ = 0.022) and

in both prebifurcation ($g$ = [3, 7]) and postbifurcation 2 ($g$ = [9, 40]). The three thalamocortical SC (i.e., woTh, Th, pTh) versions were simulated.
(TIF)

**S3 Fig. Rising cortical noise hampers $r_{PLV(\alpha)}$.** First row, showing simulations where all nodes have the same parametrization ($p$ = 0.09; $\eta$ = variable). Second row, showing simulations with the thalamus in limit cycle condition $p_{th}$ = 0.15, $\eta_{th}$ = 0.09) and a variable noisy input to cortical nodes ($p_{cx}$ = 0.09, $\eta_{cx}$ = variable).
(TIF)

**S4 Fig. Parameter space explorations for conduction speed and coupling factor in the BNM with initial parameters.** The model was parameterized as in the first in-silico experiment shown in Fig 2F (i.e., the thalamocortical experiment with high thalamic noise pth = 0.09, $\eta$th = 0.022, pcx = 0.09, $\eta$cx = 2.2e-8). Each column shows a set of simulations with a different SC version: woTh, Th, and pTh. The three heatmaps shown per column represent different measures of the same simulation including $r_{PLV(\alpha)}$, IAF as the frequency peak of the averaged spectrum from all nodes, and the power at the frequency peak of the averaged spectrum.
(TIF)

**S5 Fig. Parameter space explorations for conduction speed and coupling factor in the BNM with final parameters.** The model was parameterized following the last in-silico experiments to obtain alpha in prebifurcation as shown in Fig 7 second last column (i.e., pth = 0.15, $\eta$th = 0.09, pcx = 0.09, $\eta$cx = 2.2e-8). Each column shows a set of simulations with a different SC version: woTh, Th, and pTh. The three heatmaps shown per column represent different measures of the same simulation including $r_{PLV(\alpha)}$, IAF as the frequency peak of the averaged spectrum from all nodes, and the power at the frequency peak of the averaged spectrum.
(TIF)

**S1 Table. Network analysis of the regions included in the BNMs.** Degree, the number of neighbors of a region, and node strength, the average number of streamlines connecting a region to others were normalized over their respective maxima. Betweenness captures the number of shortest paths in a network that passes through a node. Path length stands for the average of the shortest paths for a node. Metrics were calculated with Networkx package in Python 3.9. The thalamus and the cerebellum are considered here in the parceled version. Note that Cingulate_Ant in AAL3 is divided in 3 parts and it was merged to match AAL2 scheme.
(XLSX)

## Acknowledgments

Thanks to Claudio Mirasso for his useful comments on the text, to Andrei Cristea for his support on figure design, and to the section of informatic services at UCM for the access to BRIGIT HPC cluster.

## Author Contributions

**Conceptualization:** Jesús Cabrera-Álvarez, Nina Doorn, Gianluca Susi.

**Formal analysis:** Jesús Cabrera-Álvarez.

**Investigation:** Jesús Cabrera-Álvarez, Nina Doorn.

**Methodology:** Jesús Cabrera-Álvarez.

**Software:** Jesús Cabrera-Álvarez.

**Supervision:** Gianluca Susi.

**Visualization:** Jesús Cabrera-Álvarez.

**Writing – original draft:** Jesús Cabrera-Álvarez.

**Writing – review & editing:** Jesús Cabrera-Álvarez, Nina Doorn, Fernando Maestú, Gianluca Susi.

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
