## [Decision Letter · Decision Letter 0]

4 May 2023

Dear Dr Cabrera-Álvarez,

Thank you very much for submitting your manuscript "Modeling the role of the thalamus in resting-state functional connectivity: nature or structure" for consideration at PLOS Computational Biology.

As with all papers reviewed by the journal, your manuscript was reviewed by members of the editorial board and by several independent reviewers. In light of the reviews (below this email), we would like to invite the resubmission of a significantly-revised version that takes into account the reviewers' comments.

We cannot make any decision about publication until we have seen the revised manuscript and your response to the reviewers' comments. Your revised manuscript is also likely to be sent to reviewers for further evaluation.

Sincerely,

Hayriye Cagnan

Academic Editor

PLOS Computational Biology

Daniele Marinazzo

Section Editor

PLOS Computational Biology

Reviewer's Responses to Questions

**Comments to the Authors:**

Reviewer #1: The authors construct whole-brain network models based on the JR neuronal mass, in order to study the impact of the thalamus. Even though results do not support the initial hypothesis about the importance of the thalamus, the authors still perform a thorough mechanistic exploration of the model from dynamical point of view. As a such the manuscript could be of interest to the field of computational neuroscience that studies the brain as a systems, but a more critical analysis of the reasons for the observed shortcomings is necessary.

By the construction of the brain network model, all the nodes are equal. Hence the statement that any node could play the role of the thalamus is misleading, unless it is stated that this is the case only for the way the brain model has been constructed. If the thalamus was generated from a different neuronal mass to reflect its distinctive properties, this wouldn’t have been the case. Moreover, if the paths from the thalamus were directional as it is the case in the reality, this also wouldn’t have been the case. The authors should hence refer to the attempts to model specific brain regions as separate entities in the brain network model, through multi-scale or co-simulation models, such as for example the works using TVB by Meier et al, or the works on the whole brain scaffold modeling by the lab of E. D’Angelo.

Similarly in the line 138 authors say that the thalamus did not show an impact, when it fact it did and it made the fit of the model worse. The authors should discuss why this could be the case, along the above lines, not to avoid the issue.

A second issue is the importance of time-delays for spectral activation patterns and synchronization as shown by several papers from the lab of V. Jirsa (e.g. the normalization of the connectome). Is JR model sensitive on delays? It should be at least discussed that for oscillatory activity time-delays can be of equal importance as the weights.

Within the same line, the importance of the of the graph theory metrics (and non-identifiability of the regions) appears only because of the simplicity of the model that doesn’t distinguish different brain regions, nor the directed paths from the subcortical regions, and moreover, because the model is either not sensitive to time-delays, or their impact is negligible due to the too high conduction velocity of 15m/s. Lemarechal et al Brain 2022 used the largest ever empirical cohort (close to 1000 patients) to find mean values of ~3.3 m/s, which is also the value in Petkoski & Jirsa 2022 that yields the most recognizable spectral activation patterns. So the choice of the conduction velocity needs further justification.

Minor comments:

- fig. 1 could be made for anatomically adequate.

- while rsfMRI FC is discussed as being predictive for aging, lately metrics based on dFC have been shown to have higher predictive value, see e.g. Battaglia et al Neuroimage 2020 and Petkoski et al Cereb Cortex 2023. Also adding some context in why dFC is expected to contain more information would be useful.

- discarding the tracts longer than 180mm seems way too restrictive, unless the authors refer to the Euclidian distance. But I see no reason why to use Euclidian distance instead the actual tract lengths, which could go above 250 mm.

- not clear which for frequency band are the shown PLV values.

- why the choice of 5s for the windows of dFC? is this robust? and why even a fixed length window for every band? The authors should at least try to justify their choices.

Reviewer #2: In their article, Cabrera-Alvarez et al. simulate a set of coupled Jansen-Rit models as a whole-brain network model in order to compare the model functional connectivity with empirical functional connectivity obtained with MEG data. They use structural connectivity obtained using diffusion MRI to couple the local regional models and scaling parameter g which is a classical approach in these types of models. Then they compare the model-empirical FC fit between three different versions of the local connectivity in the thalamus - namely without, with a single node and with a parcellated node thalamus - for values of g that correspond to pre- (when the models behave as damped oscillators) and post - (in limit cycle oscillators regime) bifurcation space. They also compare model-empirical FC for each case with low and high values of noise input to the thalamic node(s) that is supposed to represent afferences from the reticular system and argue that best fits are obtained in the pre-bifurcation region with high noise input to parcellated thalamus. They then use this result to claim that thalamus could be playing a driving role in the emergence of overall RS-FC.

I find that the simulations in the paper are performed with adequate detail and correctly. However, I found the rationale/motivation for the type of modelling/analysis performed lacking. I also found the interpretation of the results problematic especially with regards to the final conclusion and the title of the paper.

My first major concern is with the emphasis the authors put on the pre-bifurcation region. Looking at their results in this region, the model-empirical FC fit for parcellated thalamus case with high noise is significantly higher than without thalamus & with single node thalamus models. However, the rPLV values in the post-bifurcation region are in the same range as these highest pre-bifurcation values (pTh + high noise). Therefore it’s not clear why the authors focussed on the pre-bifurcation region specifically later on, i.e. what additional biological/empirical evidence (power spectrum for instance?) implies that the pre- and not post-bifurcation region is the optimal one is not clear.

Instead of using global FC metrics, authors could consider specific FCs and see how they match between the model & empirical data. For instance which model - parcellated or single-node thalamus - gives a better matching empirical thalamocortical FC could be explored to fine-tune both the parameter-space (g) as well as the choice of pre-vs-post bifurcation regime.

Authors show that instead of the thalamus, using a parcellated cerebellum gives the same levels of model-empirical FC fits as with the thalamus. They use this result to say that the specific structure in the thalamus does not matter and it’s the noise. But the flip side of this observation is also that within the limited confines of the metrics (FC and dFC similarity) used by the authors and the model, thalamus is not unique. So how could one make a claim, based purely on these observations, that the thalamus is performing a driving role in emergence of FC? Perhaps it’s the cerebellum? Either the authors should moderate their claims about the role of the thalamus in the article or they should use additional metrics (perhaps local FC instead of global, brain-wide FC ?) to make a distinction between the thalamus vs cerebellum comparisons. Also, I was curious to see what happens if both the cerebellum and thalamus are parcellated (that would be closer to biological reality anyway) ?

It is not clear how the authors chose the noise levels. There should be an explanation for the values chosen by the authors.

The readability of the manuscript can be improved is significantly overall. Sub-figures can be labelled A, B etc. Figure captions at the moment are extremely inadequate and they can’t be understood without going through the results section. That should not be the case.

I take it that the horizontal blue lines in the FFT plots in figures 5 and 6 represent bifurcation thresholds only? I would suggest to use a different color that’s not included in the colormaps of these figures for the lines in that case.

**Have the authors made all data and (if applicable) computational code underlying the findings in their manuscript fully available?**

Reviewer #1: None

Reviewer #2: Yes

PLOS authors have the option to publish the peer review history of their article (what does this mean?). If published, this will include your full peer review and any attached files.

Reviewer #1: No

Reviewer #2: **Yes: **MOHIT H ADHIKARI
---

## [Decision Letter · Decision Letter 1]

10 Jul 2023

Dear authors,

We are pleased to inform you that your manuscript 'Modeling the role of the thalamus in resting-state functional connectivity: nature or structure' has been provisionally accepted for publication in PLOS Computational Biology.

Please ensure to clarify data availability at the proof stage.

Best regards,

Hayriye Cagnan

Academic Editor

PLOS Computational Biology

Daniele Marinazzo

Section Editor

PLOS Computational Biology

Reviewer's Responses to Questions

**Comments to the Authors:**

Reviewer #1: The authors have addressed all of my comments

Reviewer #2: The authors have sufficiently addressed my comments.

**Have the authors made all data and (if applicable) computational code underlying the findings in their manuscript fully available?**

Reviewer #1: None

Reviewer #2: None

PLOS authors have the option to publish the peer review history of their article (what does this mean?). If published, this will include your full peer review and any attached files.

Reviewer #1: No

Reviewer #2: **Yes: **Mohit Adhikari

---

## [Editor Report · Acceptance letter]

26 Jul 2023

PCOMPBIOL-D-23-00363R1 

Modeling the role of the thalamus in resting-state functional connectivity: nature or structure

Dear Dr Cabrera-Álvarez,

I am pleased to inform you that your manuscript has been formally accepted for publication in PLOS Computational Biology. Your manuscript is now with our production department and you will be notified of the publication date in due course.

With kind regards,

Zsofi Zombor
